# GRB2 stabilizes RAD51 at reversed replication forks suppressing genomic instability and innate immunity against cancer

Zu Ye[1,7], Shengfeng Xu[2], Yin Shi[3,4], Xueqian Cheng[5], Yuan Zhang[1], Sunetra Roy[6], Sarita Namjoshi[1], Michael A. Longo[1], Todd M. Link[1], Katharina Schlacher [6], Guang Peng [5], Dihua Yu [1], Bin Wang [2], John A. Tainer [1] ✉ & Zamal Ahmed [1] ✉

Growth factor receptor-bound protein 2 (GRB2) is a cytoplasmic adapter for tyrosine kinase signaling and a nuclear adapter for homology-directed-DNA repair. Here we find nuclear GRB2 protects DNA at stalled replication forks from MRE11-mediated degradation in the BRCA2 replication fork protection axis. Mechanistically, GRB2 binds and inhibits RAD51 ATPase activity to stabilize RAD51 on stalled replication forks. In GRB2-depleted cells, PARP inhibitor (PARPi) treatment releases DNA fragments from stalled forks into the cytoplasm that activate the cGAS–STING pathway to trigger pro-inflammatory cytokine production. Moreover in a syngeneic mouse metastatic ovarian cancer model, GRB2 depletion in the context of PARPi treatment reduced tumor burden and enabled high survival consistent with immune suppression of cancer growth. Collective findings unveil GRB2 function and mechanism for fork protection in the BRCA2-RAD51-MRE11 axis and suggest GRB2 as a potential therapeutic target and an enabling predictive biomarker for patient selection for PARPi and immunotherapy combination.

GRB2 is an intracellular adapter protein essential for cell proliferation that consists of a central SH2 domain flanked by two SH3 domains[1–5]. Classically its SH3 domains direct complex formation with proline-rich regions of other proteins, and its SH2 domain binds tyrosine phosphorylated sequences[6–10]. Cytoplasmic GRB2 acts in initial steps of receptor tyrosine kinase (RTK) signaling to the Ras-MAPK cascade. In the nucleus GRB2 adapter moonlights in initial steps for efficient homology-directed-DNA of DNA double strand breaks (DSBs)[11]. Given these dual functions, we reasoned that GRB2 could act more generally in the DNA damage response (DDR) including its DNA replication stress responses and the activation of the innate immune response by loss of genome stability during replication. In particular, tumor cells with oncogenic replication stress might select for GRB2 activities associated with proliferation and the DDR. Yet GRB2 nuclear activities and mechanisms as well as their possible connections to cancer are largely undefined.

[1]Departments of Molecular and Cellular Oncology and Cancer Biology, The University of Texas MD Anderson Cancer Center, Houston, TX 77030, USA. [2]Department of Genetics, The University of Texas MD Anderson Cancer Center, Houston, TX 77030, USA. [3]Department of Biochemistry, Zhejiang University School of Medicine, Hangzhou 310058, China. [4]Division of Pediatrics, The University of Texas MD Anderson Cancer Center, Houston, TX 77030, USA. [5]Department of Clinical Cancer Prevention, The University of Texas MD Anderson Cancer Center, Houston, TX 77030, USA. [6]Department of Cancer Biology, University of Texas MD Anderson Cancer Center, Houston, TX 77030, USA. [7]Present address: Zhejiang Cancer Hospital, Hangzhou Institute of Medicine (HIM), Chinese Academy of Sciences, Hangzhou, Zhejiang 310022, China. ✉e-mail: JTainer@mdanderson.org; ZAhmed@mdanderson.org

Here we tested and defined GRB2 interactions and activities with DNA replication fork proteins in vitro, in cells, and in vivo. We find that GRB2-depleted cells strikingly mirrored DNA replication fork protection characteristics of the BRCA2-deficient phenotype. In response to replication stress from hydroxyurea (HU) and PARP inhibitor (PARPi), GRB2 stabilized RAD51 on single-stranded DNA (ssDNA) at stalled replication forks to reduce cytoplasmic DNA accumulation and cGAS/STING activation. Furthermore, BRCA2 knockdown (KD) in a GRB2-KO background showed no added fork degradation defects consistent with an epistatic relationship. During RF stress, GRB2 depletion promoted cGAS/STING activation that translated into the production of inflammatory cytokines and recruitment of cytotoxic T-cells. In a GRB2-depleted syngeneic ovarian cancer mouse model, PARPi treatment led to enhanced targeted destruction of tumor cells by the host immune system compared to PARPi alone. These findings align GRB2 with BRCA2 for replication fork protection and PARPi responses that include the therapeutic activation of innate immunity.

## Results

### GRB2 binds DNA replication forks and functions in the replication stress response

GRB2 in the cytoplasm acts in growth factor receptor tyrosine kinase (RTK) and Ras/MAP kinase activation driving *quiescent* ($G_0$) cells to enter cell-cycle. Consistent with these observations, our previously generated GRB2 knockdown (GRB2-KD) in A431 cells[12] showed a measurable G1-phase stagnation and reduced S-phase duration compared with wild type (control) without affecting overall cell viability. We therefore asked whether nuclear GRB2 may impact efficient S-phase progression. We treated cells with low dose hydroxyurea (HU) to create replication stress and found that GRB2-KD cells showed a dramatic increase in S-phase stagnation within a 24 h time period (Supplementary Fig. 1a). GRB2-KD cells, challenged with HU, required a longer time to complete DNA duplication implying GRB2 acts in maintaining S-phase efficiency under replication stress. Similar results were observed in GRB2 knockout (KO) HAP-1 cells (Supplementary Fig. 1b). A GRB2 functional role was also shown in colony survivals assay where HU and mitomycin C (MMC) treatment severely impeded colony formation in GRB2-KO cells that can however be rescued by reconstitution of GRB2 (Supplementary Fig. 1c, d).

Mass spectrometry previously identified proliferating cell nuclear antigen (PCNA) as a nuclear GRB2 interacting protein[11]. PCNA a trimeric docking protein for recruitment of replication fork (RF) proteins including DNA polymerases, FEN1 nuclease, and DNA ligase[13]. We therefore investigated GRB2 at the RF using fluorescence lifetime imaging microscopy (FLIM) to measure fluorescence resonance energy transfer (FRET) between GRB2 and PCNA. Under normal conditions GRB2 co-localized with PCNA, but overlap was difficult to assess (Fig. 1a, left panel and Supplementary Fig. 2a). Yet, FLIM showed a measurable FRET between PCNA and GRB2 indicated by the reduction in the average lifetime of the GFP-tagged PCNA (Fig. 1a, right panel and Supplementary Fig. 2b). mEmerald-PCNA overexpressed cells showed some cytoplasmic localization when co-expressed with RFP-tagged GRB2. Investigation of potential GRB2-meditated PCNA retention in the cytoplasm found that cytoplasmic-PCNA localization is likely an overexpression artifact (Supplementary Fig. 2c). Indeed, FLIM accurately measures interaction between GRB2 and PCNA inside the nucleus circumventing contrast-based imaging localization artifacts (Supplementary Fig. 2a, b). The direct interaction between PCNA and GRB2 was further supported by an in vitro binding assay using purified proteins (Supplementary Fig. 2c, d). We then identified residue 200-209 (QTGMFPRNY) as an embedded putative PCNA interaction peptide (PIP) motif within the C-terminus of GRB2. We found that synthetic GQTGMFPRNY peptide bound PCNA with a dissociation constant comparable to $^{WT}$GRB2 (Supplementary Fig. 1c). As PCNA is tyrosine phosphorylated on residue Y211[14–16] to form a potential pYxNx

GRB2 SH2 domain binding site, we tested binding and found that tyrosine phosphorylated PCNA (pPCNA) binds GRB2 with an order of a magnitude higher affinity in the MST binding assay (Supplementary Fig. 2c). Given the close association of PCNA with the DNA replication fork (RF), these collective data supported GRB2 interactions and function at the RF.

We therefore investigated GRB2's direct association with the RF by using the isolation of Proteins On Nascent DNA (iPOND) method[17,18]. This assay was performed in cells unperturbed or challenged with HU-induced replication stress. As expected, GRB2 was present in the progressing RFs marked with PCNA accumulation. Interestingly, more GRB2 was loaded onto the stalled RF, despite the apparent PCNA dissociation under HU treatment, suggesting other potential GRB2 interactions at the RF (Fig. 1b). This enrichment of GRB2 on nascent DNA at stalled replication forks was validated by the quantitative in situ analysis of protein interactions at DNA replication forks (SIRF) assay (Supplementary Fig. 3)[19].

We further investigated the GRB2 function at RFs with DNA-fiber assay measurements of replication fork progression between control and GRB-KO cells (Supplementary Fig. 4a–c). IdU track length measurements under normal cell growth conditions found no significant difference between the WT and GRB2-KO HeLa and HAP-1 cells (Fig. 1c and Supplementary Fig. 4c,d). However, with replication stress, a significant reduction in IdU stained DNA tract led to decreased IdU/CIdU ratio in GRB2-KO cells (Fig. 1c and Supplementary Fig. 4d). Importantly, GRB2-KO cells reconstituted with exogenous GRB2 restored replication progression to levels comparable to wild-type parental cells (Fig. 1d). These data show GRB2 localizes to RFs and functions in the replication stress response.

### GRB2 protects stalled replication forks from MRE11-mediated degradation

As mutant GRB2$^{K109R}$ disrupts the GRB2-MRE11 interaction for DNA double-strand break repair while retaining RTK activity[11], we employed the impact of this separation-of-function mutant on stalled RFs. We found that reconstitution with GRB2$^{K109R}$ failed to rescue fork degradation in GRB2-KO cells (Fig. 1d). This finding is consistent with RF degradation by MRE11 in GRB2-KO cells and suggested GRB2 interaction with MRE11 can restore RF protection.

Unprotected stalled RFs can undergo extensive nucleolytic processing generating single stranded (ss) DNA[20,21]. We therefore employed immunofluorescence analysis with an antibody to single-stranded (ss) DNA (ssDNA). We measured a significant increase in ssDNA accumulation in GRB2-KO cells treated with HU compared to control cells (Figs. 1e, f). We then investigated if the observed reduction in IdU stained DNA tract and the accumulation of ssDNA was the result of MRE11 nuclease processing of stalled RFs. Specifically, we employed the single-molecule DNA fiber assay to investigate the effect of MRE11 inhibitor Mirin on HU-induced fork degradation. Mirin treatment had no effect on the integrity of stalled RFs in the control HeLa or HAP1 cells, while in GRB2-KO cells HU-induced fork degradation was prevented (Fig. 1g and Supplementary Fig. 4e). The specificity of MRE11-mediated stalled replication fork processing was further confirmed by MRE11-knockdown (KD), which like Mirin rescued fork degradation in GRB2-KO HeLa cells (Fig. 1h and Supplementary Fig. 5a). We also examined other RF processing nucleases, EXO1 and DNA2. Knockdown of EXO1 or DNA2 failed to rescue fork degradation in GRB2-KO cells (Supplementary Fig. 5b).

These collective data indicate GRB2-MRE11 interaction blocks excessive processing of stalled replication forks to limit generation of ssDNA. This is supported by the findings that GRB2-MRE11 interaction disrupting $^{K109R}$GRB2 mutant failed to prevent MRE11-mediated stalled RF degradation (Fig. 1d) or to restore cell replication stress resistance induced by HU or MMC treatment in colony survival assays (Supplementary Fig. 1c).

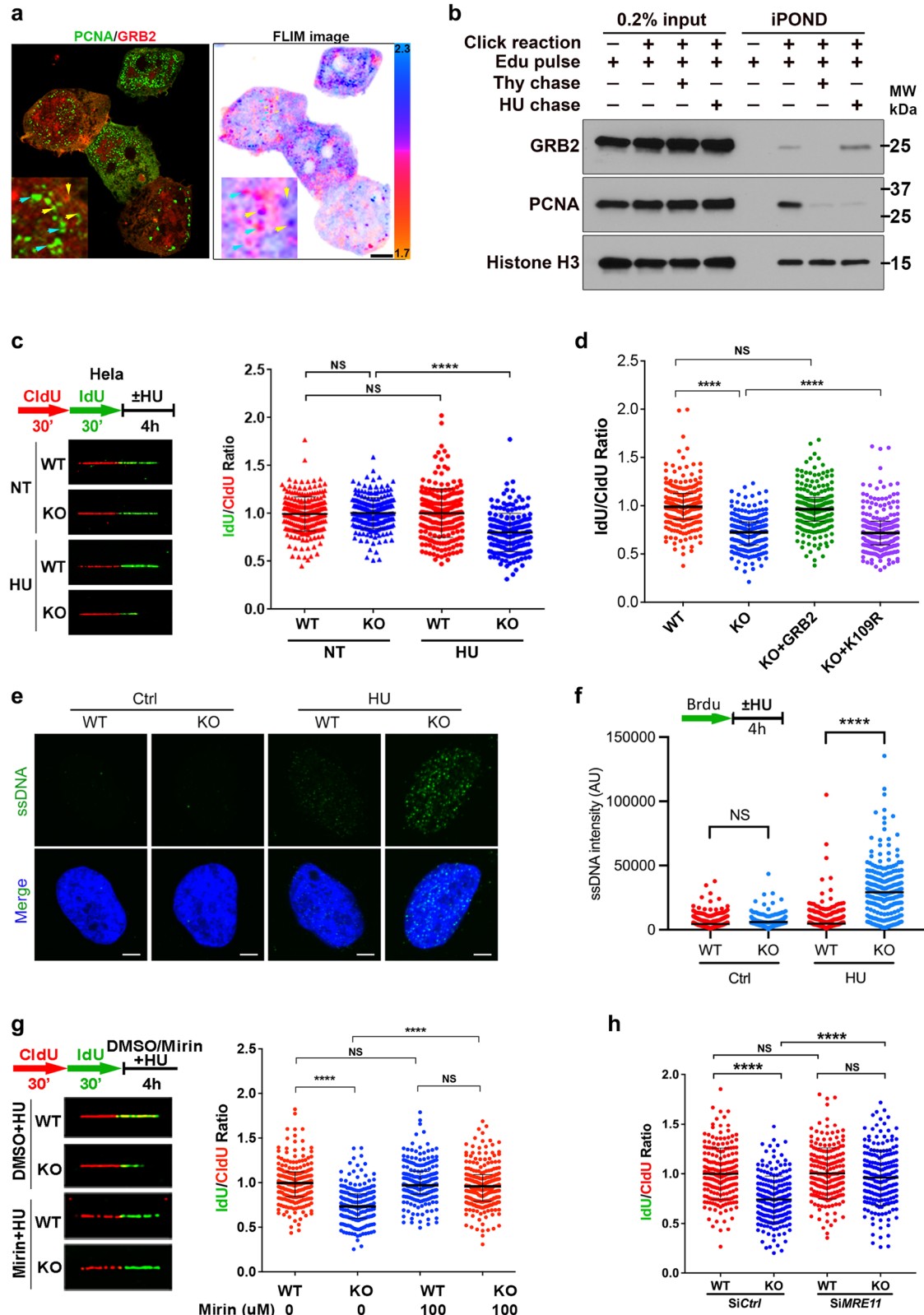

## GRB2 depletion mirrors BRCA deficiency in loss of replication fork protection

In BRCA deficient cells, MRE11-dependent fork degradation creates long stretches of ssDNA, as one of the main causes of PARP inhibitor sensitivity[22]. Our GRB2 KO HeLa and HAP-1 cells express BRCA2[23,24], yet GRB2-KO induces RF instability and fork degradation (Fig. 1c–h). GRB2-depleted cells show high susceptibility to MRE11 nuclease

mediated degradation upon replication stress. In the absence of GRB2, RFs lack protection against the MRE11 nuclease – mirroring BRCA deficiency and implying GRB2 acts in parallel or in conjunction with the BRCA fork protection axis. MRE11-mediated fork degradation is suppressed by BRCA2-mediated RAD51 nucleoprotein filament stabilization in wild type cells[25–27]. BRC4 fragment of BRCA2 blocks nucleoprotein filament formation by binding to RAD51 monomers[28].

**Fig. 1 | GRB2 at the DNA replication fork inhibits MRE11 mediated fork-degradation. a** FLIM/FRET showing GRB2-PCNA direct interaction within an apparent PCNA replication focus. Colocalization shown in the left, while false colored lifetime image on the right generated by pixel-by-pixel mapping of the measured lifetime-values represented by the scale 1.7–2.2 nanoseconds. Scale bar 25 μm (**b**) iPOND assay showing GRB2 association with replication DNA, and enriched on the nascent DNA in response replication stress. PCNA was used as a positive control. **c** Enhanced fork degradation in GRB2-depleted HeLa cells under replication stress. **d** An independent set of experiments with GRB2 reconstitution. Only ᵂᵀGRB2 (KO + GRB2), but not the K109R mutant alleviated replication stress induced fork degradation in GRB2-depleted HeLa cells. Radiometric analysis of

≥200 fibers, $n = 3$. **e** Replication stress induced increased ssDNA in GRB2-depleted cells under replication stress indicating enhanced nuclease activity in HeLa cells. Scale bar 10 μm. **f** Quantitation of the ssDNA intensities from three independent experiments represented in (**e**), intensities of 100–300 foci, $n = 3$. **g** Fork degradation is inhibited by MRE11 nuclease inhibitor Mirin in HeLa cells lacking GRB2 and under replication stress. **h** MRE11knockddown prevents fork degradation in GRB2-depleted HeLa cells under replication stress. Fiber assay radiometric analysis of ≥ 200 fibers, $n = 3$ in (**c**, **d**, **f**–**h**). The significance was analyzed by two-sided Student's $t$ test. ***$P \le 0.001$, and ****$P \le 0.0001$; NS not significant. Error bars showing standard deviations (SD). For (**b**–**d**) and (**f**–**h**), source data are provided as a Source Data file.

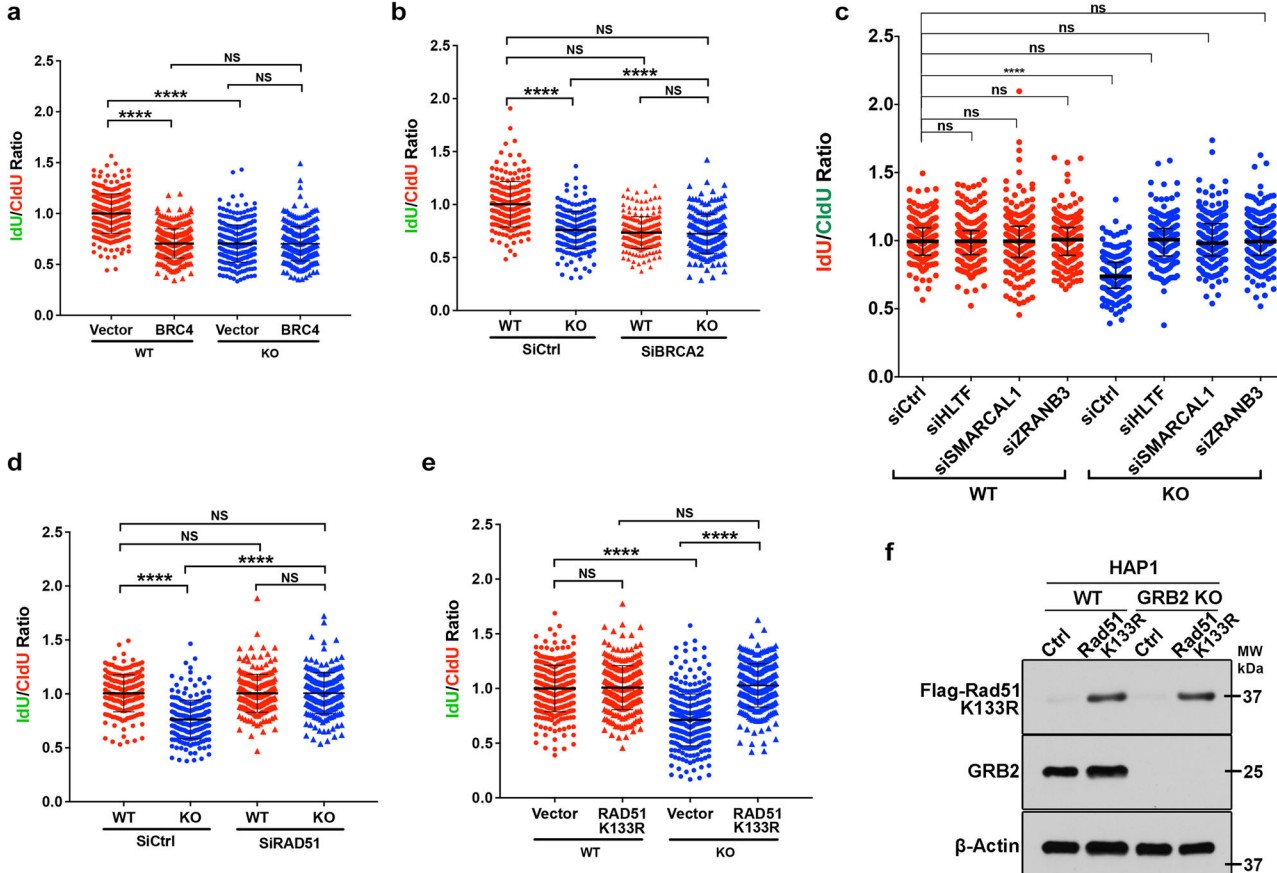

**Fig. 2 | Inhibition of fork reversal rescues fork degradation in GRB2 deficient cells. a** Overexpression of BRC4 has no further fork degradation effect on GRB2 deficient cells. **b** BRCA2 knockdown has no further fork degradation effect on GRB2 deficient cells. **c** Knockdown of HTLF, SMARCAL1 or ZRANB3 is sufficient to alleviate replication fork degradation in GRB2-depleted cells. **d** Knockdown of RAD51 rescues the observed fork degradation in GRB2-depleted cells. **e** Overexpression of K133R ATPase deficient mutant RAD51 is sufficient to prevent fork degradation in GRB2-depleted HAP1 cells. **f** Western blot analysis

showing the expression of ᴷ¹³³ᴿRAD51 mutant in WT and GRB2 KO HAP1 cells using anti-Flag antibody. Antibody targeting GRB2 was used to distinguish WT from GRB2 KO cells. Beta-Actin serves as a loading control. In (**a**–**e**), radiometric analysis of ≥200 fibers, $n = 3$. The significance was analyzed by two-sided Student's $t$ test. *$P \le 0.05$, **$P \le 0.01$, ***$P \le 0.001$, and ****$P \le 0.0001$; NS, not significant. Error bars showing standard deviations (SD). For (**a**–**f**), source data are provided as a Source Data file.

We therefore investigated GRB2 fork protection with respect to the BRCA2 fork protection.

Overexpressing of BRC4 fragment acts as an interfering, dominant negative in BRCA2 expressing cells to cause fork defects[25], so we tested the effect of BRC4 expression on control and GRB2-KO cells. As expected, BRC4 expression in the control cells showed a measurable effect on fork degradation. However, BRC4 overexpression in GRB2-KO cells showed no aggravated fork degradation (Fig. 2a and Supplementary Fig. 6a). To further understand the role of BRCA2, we created BRCA2-KD through siRNA transfection in control and GRB2-KO HeLa cells (Supplementary Fig. 6b). In control cells, BRCA2-KD caused RF

degradation comparable to GRB2-KO, whereas BRCA2-KD in GRB2-KO background showed no added fork degradation defect (Fig. 2b). This observation suggests that GRB2 fork protection may function upstream to BRCA2 or a limitation of the DNA fiber assay in detecting RF degradation beyond a certain length.

Inhibition of RF reversal leads to fork protection in GRB2-deficient cells. Reversed forks are highly susceptible to cellular nucleases in the absence of BRCA1/2[27,29,30]. We therefore tested if fork degradation occurs at the stalled or reversed replication forks. Indeed, BRCA2 protects reversed forks from MRE11-mediated degradation[25]. The SNF2 family proteins, including SMARCAL1, HLTF and ZRANB3 are

important fork reversal enzymes[31–33]. To test whether MRE11-mediated RF degradation in GRB2-KO cells depends on fork reversal, we individually knocked down fork reversal enzymes HLTF, ZRANB3 and SMARCAL1 in control and GRB2-KO cells (Supplementary Fig. 6c–e). In control cells, knockdown of HLTF, ZRANB3 or SMARCAL1 showed no measurable impact on replication fork degradation. However, knockdown of HLTF, ZRANB3 or SMARCAL1 in GRB2-KO cells rescued RF degradation (Fig. 2c). Notably SMARCAL1, ZRANB3 and HLTF function independently at RFs[34]. This suggests that GRB2 functions at the reversed replication forks and that reduced fork-reversal can mitigate the GRB2-KO effect. In both prokaryotes and eukaryotes RAD51 is also required for fork reversal that is independent of BRCA2[27,35]. To gain further insight into GRB2's role in fork stability, we knocked down RAD51 in control and GRB2-KO cells and again performed DNA fiber assays. No measurable differences were seen in the control cells but RAD51-KD was sufficient to rescue RF protection defect of GRB2-KO cells (Fig. 2d and Supplementary Fig. 6f). Experimental results revealed RAD51-KD, like knockdown of SMARCAL1, ZRANB3 or HLTF, rescued fork degradation induced by GRB2-KO under HU treatment. Thus, collective results indicate GRB2 prevents over-resection of MRE11 nuclease at reversed RFs.

RAD51 on ssDNA is required for both RF reversal and RF protection. In BRCA2-deficient cells, overexpression of an ATP-hydrolysis deficient RAD51 mutant K133R prevented MRE11-dependent fork degradation[27,36]. We therefore tested the effect of RAD51 ATPase activity relevant in RF degradation in GRB2-KO cells. Overexpression of ATPase defective K113R mutant in GRB2-KO cells rescued fork degradation (Fig. 2e, f). These collective results indicate GRB2 acts in RAD51-mediated protection of reversed RFs and supports a GRB2 function beyond its MRE11 binding and recruitment.

## GRB2 binds and inhibits RAD51 ATPase to stabilize RAD51-DNA complex at stalled RFs

Classically, BRCA2 is required for RAD51-mediated DNA filament formation to prevent MRE11 mediated degradation of reversed RFs[20]. In BRCA2-deficient cells, overexpression of RAD51 K133R, an ATP-hydrolysis deficient mutant, prevented MRE11-dependent fork degradation[25,27,36,37]. A RAD51 gradient controlling stalled RFs has been proposed where a small number of RAD51 monomers are required for fork reversal, a higher concentration for fork protection and restart requiring long RAD51 filaments[38]. To investigate if dysregulated MRE11 nuclease activity in GRB2-KO cells is due to insufficient RAD51 protection, RAD51 K133R was overexpressed to form a stable RAD51 nucleofilament. This ATP-hydrolysis defective mutant is expected to remain bound to ssDNA and dsDNA, and overexpression of RAD51 K133R was sufficient to rescue fork defects caused by GRB2-KO (Fig. 2e, f).

In an immunoprecipitation assay GRB2 was seen to form complexes with PTEN and RAD51 in response hydrogen peroxide treatment[39]. We therefore tested for direct RAD51 binding. We found that GRB2 directly binds RAD51 with tight 300 nM dissociation constant (Kd) (Fig. 3a). An in vitro GST pulldown assay with purified individual GRB2 SH-domains identified a GRB2-SH2 domain interaction with RAD51 (Supplementary Fig. 6g). The binding Kd of isolated GRB2-SH2 domain was comparable to that seen with full-length protein. Since the GRB2 SH2 domain interacted with EGFR and MRE11 in a distinct binding surface[11], we also tested if R86A and K109R GRB2 mutants, which disrupt EGFR and MRE11 binding respectively, impact RAD51 interaction. The results showed the interaction of RAD51 with R86A was comparable to the WT while the MRE11-interaction disrupting mutant K109R showed a measurable decrease in RAD51 binding affinity (Fig. 3a). The K109A GRB2 mutant that binds but doesn't release MRE11[11], also retains RAD51 binding at an affinity comparable to the ᵂᵀGRB2. Our affinity measurements suggest an overlapping interaction surface on GRB2-SH2 domain shared by both MRE11 and RAD51

consistent with a handoff from MRE11 to RAD51 or with distinct complexes.

ATP hydrolysis releases RAD51 from ssDNA[40,41], and ATPase-deficient RAD51 mutant protects RFs from degradation in GRB2-KO cells. We therefore asked if GRB2 could protect RFs by inhibiting RAD51 and tested the effect of GRB2-RAD51 interaction on RAD51 ATPase activity. GRB2 was titrated into a fixed concentration of RAD51, and ATPase activity was measured. The results showed a dose-dependent inhibition of RAD51 ATPase activity with increasing concentration of GRB2 (Fig. 3b). The inhibition of RAD51 ATP hydrolysis in vitro was shown to promote stable strand exchange and formation of a stable complex with dsDNA with a 5′ overhang[41]. We therefore mixed a 32nt duplex DNA with one strand fluorescently labeled with a 90nt unlabeled ssDNA containing complementary sequence to the labeled 32nt and monitored the exchange of the labeled 32nt. The results showed that GRB2, in a dose-dependent manner, induced an efficient RAD51-mediated exchange of labeled 32nt with the complementary 90nt sequence (Fig. 3c, d).

To further test the impact of GRB2 on RAD51 stability at the replication fork, we used the proximity ligation SIRF assay[19] to measure RAD51 proximity to nascent ssDNA. In cells under normal growth condition, there was no significant difference in RAD51 localization between the control (WT) and GRB2-KO (KO) cells (Fig. 3e). In WT cells, HU treatment induced a significant increase in RAD51-bound nascent DNA foci compared to GRB2-KO cells. HU treated GRB2-KO cells showed a significantly lower level of DNA-bound RAD51 (Fig. 3e, f), while the expression pattern of RAD51 was not affected by the GRB2 Knockout (Supplementary Fig. 6f). These data are consistent with in vitro observations where higher RAD51 ATPase activity correlated with decreased strand stabilization to 5′-overhang dsDNA without GRB2 (Fig. 3b–d). The RAD51 K133R mutant is ATP-hydrolysis deficient[25,27,36,37]. To further understand the impact of GRB2 on RAD51 ATPase activity and replication fork stability, we performed dose-escalation overexpression analysis of the ATPase-deficient K133R-RAD51 in GRB2-KO cells. In control cells with normal levels of GRB2, K133R expression level had little impact. However, a 2-fold higher overexpression of K133R in GRB2-KO cells was sufficient to rescue PARPi sensitivity to a level comparable to the parental control cells (Supplementary Fig. 6h, i). Thus, GRB2 evidently protects RFs by inhibiting RAD51 ATPase activity to maintain stable RAD51-DNA interactions. We envisage that in the absence of GRB2, ATP hydrolysis of RAD51 loaded onto reversed fork leads to its premature dissociation from DNA, leaving them vulnerable to MRE11-dependent nucleolytic processing.

## GRB2 restricts cGAS/STING activation and inflammatory cytokine release under replication stress

In general, PARPi sensitivity in BRCA1/2 defective cells and tumors correlates with fork protection defects that can generate cytosolic ssDNA and trigger cGAS/STING activation[21,42–44]. We therefore examined cells with GRB2 deficiency and found that they are vulnerable to MRE11-mediated degradation of reversed RFs, and therefore sensitive to replication stress induced by HU or MMC treatment (Supplementary Fig. 1c, d). GRB2-KO cells are also sensitive to PARPi[11], and PARPi sensitivity induces cytoplasmic micronuclei formation[26,45,46]. We therefore tested and found PARPi (Olaparib) treatment induced a significantly higher level of cytoplasmic micronuclei in GRB2-depleted HAP-1 and HeLa cells compared to controls (Fig. 4a and Supplementary Fig. 7a, b). Notably, reconstitution of GRB2 in GRB2-KO cells suppressed Olaparib-induced micronuclei (Fig. 4b and Supplementary Fig. 7a, b) underscoring the importance of GRB2 levels in control of micronuclei.

Production of excessive micronuclei in PARPi-treated GRB2-KO cells activated the cGAS/STING pathway, as measured by increased phosphorylated TBK1 (p-TBK1) and its downstream target IRF3

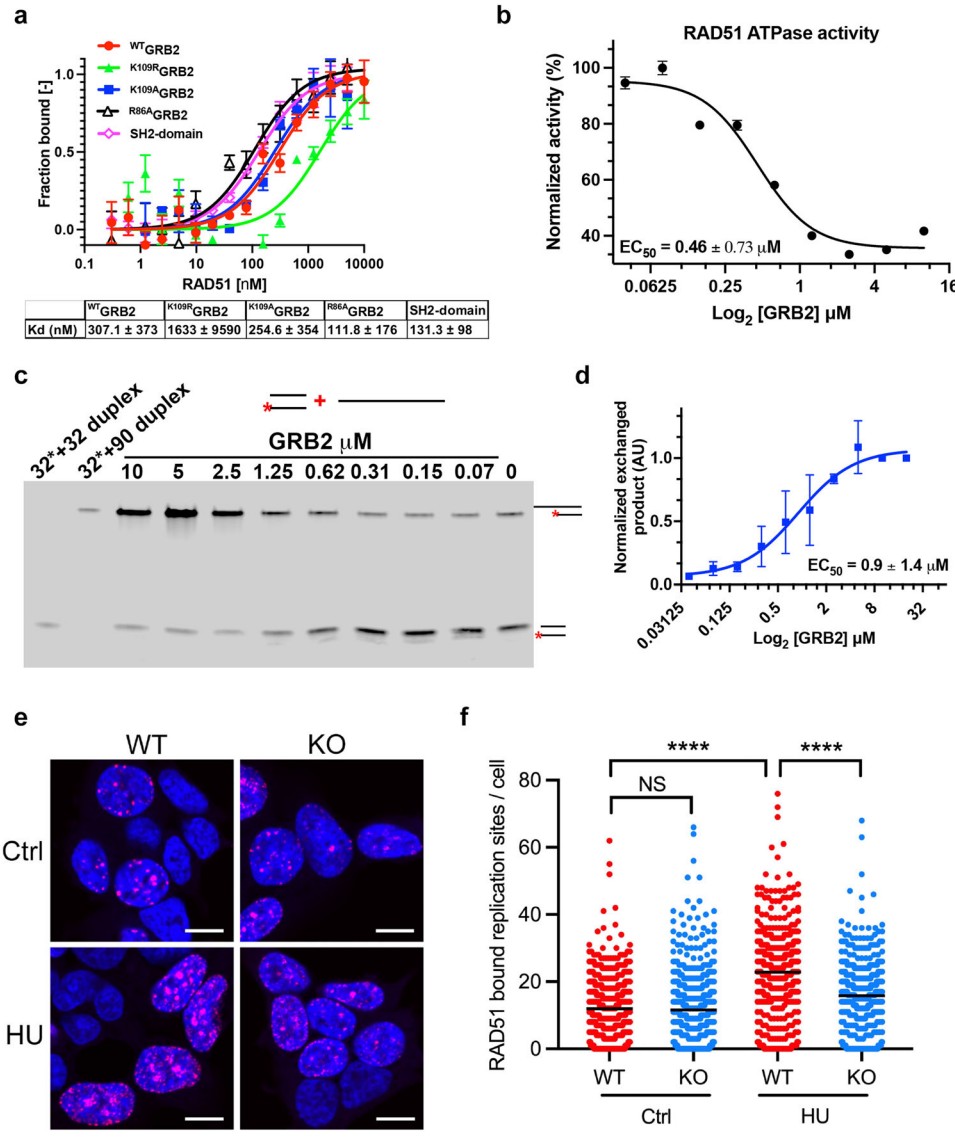

**Fig. 3 | GRB2 promotes reverse fork stability by inhibiting RAD51 ATPase activity. a** RAD51 interacts with the SH2 domain of GRB2. MST binding isotherms measuring binding affinity of RAD51 with wild type GRB2, SH2 domain and indicated GRB2 mutant. The binding affinities (Kds) mean values with ±SD are shown below, $n = 3$. **b** ATPase activity assay showing GRB2 dose-dependent inhibition of RAD51 ATP-hydrolysis with ±SD, $n = 3$. **c** Representative gel-image showing GRB2 dose dependent RAD51 strand-stabilization to dsDNA with 5′ overhang. **d** Normalized quantitation of RAD51 strand exchange and stability induced by GRB2 with ±SD, $n = 3$. **e** Representative PLA images of RAD51 foci formation with ssDNA. **f** Quantitation of nascent DNA associated RAD51 collected from >350, $n = 3$. Scale bar 10 μm. The significance was analyzed by two-sided Student's $t$ test. ****$P ≤ 0.0001$; NS not significant. For **a**–**d** and **f** source data are provided as a Source Data file.

phosphorylation (pIRF3) levels. Both p-TBK1 and pIRF3 were measurably upregulated in PARPi treated GRB2-KO cells in all tested time periods (Fig. 4c and Supplementary Fig. 7c, d). Importantly, reconstitution of GRB2 in GRB2-KO cells restored p-TBK1 and p-IRF3 level comparable to the control (Fig. 4d and Supplementary Fig. 7e, f). These data imply GRB2 expression alone is sufficient to restrict cytoplasmic DNA accumulation.

Under normal conditions IRF-3 resides in the cytoplasm of cells but phosphorylation triggered by cytoplasmic DNA leads to its translocation to the nucleus. This phosphorylation and translocation in association with the p300/CBP coactivator protein promotes DNA binding and transcriptional activity[47]. We therefore sought to determine if the functionally important p-IRF3 nuclear translocation occurred. Immunofluorescence analysis revealed a drastic increase in pIRF3 level in the nucleus of GRB2-KO cells treated with PARPi (Fig. 4e). These data are consistent with western blot analysis and further imply

increased transcriptional activity of IRF3 in GRB2-KO cells in response to PARPi treatment.

As GRB2 acts in the RAS and PI3-Kinase signaling pathway[48], we tested to see if the observed cGAS/STING upregulation is due to its effect on cytoplasmic signaling. Control and GRB2-KO cells were treated with MEK and AKT inhibitors and the pTBK1 and pIRF3 levels were measured. The results showed that inhibition of RAS-MAPK or PI3-K signaling pathways have little effect on cGAS/STING signaling (Fig. 4f and Supplementary Fig. 7g, h). Thus, the pTBK1 response observed here corresponds to GRB2 function in DNA replication stress.

In GRB2-depleted cells, MRE11 resected stalled replication forks (Fig. 1g, h). We therefore tested if the observed pTBK1 and pIRF3 upregulation in GRB2-KO cells correlates with MRE11 expression level. We knocked down MRE11 (siMRE11) in GRB2-KO cells and measured TBK1 and IRF3 phosphorylation. The depletion of MRE11 in GRB2-KO cells inhibited PARPi induced pTBK and pIRF3 phosphorylation (Fig. 4g

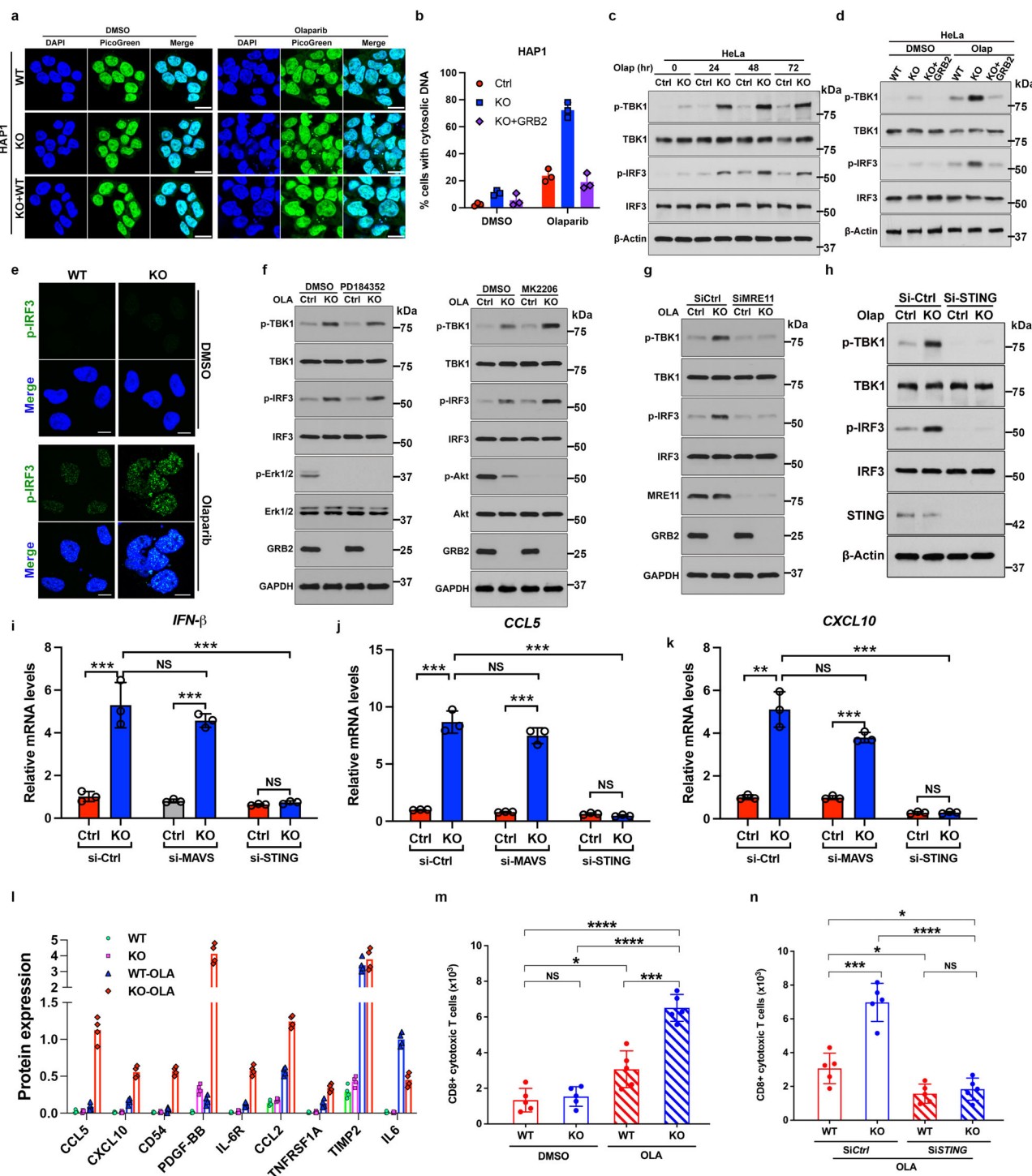

and Supplementary Fig. 7i). These results suggest that the elevated TBK1 and TRF3 phosphorylation in GRB2-KO cells is caused by cytoplasmic DNA generated by MRE11 nucleolytic processing in response to PARPi-induced replication stress. To test if inhibiting MRE11 nuclease activity can also inhibit PARPi-induced pTBK1 in GRB2-KO cells, we treated control and GRB2-KO cells with MRE11 inhibitors Mirin with and without olaparib. Our results showed MRE11 inhibitors did not fully recapitulate the MRE11-KD phenotype in GRB2-KO cells (Supplementary Fig. 7j–l). Although there was a measurable pTBK1 reduction in the combined MRE11i and PARPi treated cells, it was not as effective as MRE11-KD. Thus, MRE11i may either lack sufficient potency to provide full protection against degradation of an unprotected replication

fork without GRB2 being present or MRE11i promotes alternative activities at forks compared to MRE11-KD in the absence of GRB2.

TBK1 and IRF3 phosphorylation can be upregulated by STING or MAVS[49,50]. To elucidate which pathway activation resulted in TBK1 and IRF3 phosphorylation, we used shRNA knock down of either STING or MAVS to show that the cGAS/STING, but not the MAVS, signaling pathway is activated in GRB2-KO cells treated with PARPi (Fig. 4h and Supplementary Fig. 8a–c).

We next tested and found cGAS/STING activation correlated with upregulation of type I IFNs and pro-inflammatory cytokine mRNA. In GRB2-KO cells, IFNβ, CCL2 and CXCL10 mRNA were upregulated in response to PARPi treatment (Fig. 4i, k). To determine if the observed

**Fig. 4 | PARPi induced DNA replication stress in GRB2 depleted cells cause genomic instability, cytoplasmic DNA accumulation resulting in cGAS/STING activation and chemoattractant secretions. a** Representative PicoGreen and DAPI staining after 10 μM Olaparib treatment for 48 h and the resulting cytoplasmic micronuclei formation in HAP-1 cells. Scale bar 10 μm (**b**) Quantitation of cytoplasmic micronuclei form three independent experiments with ±SD. **c** A comparison and time-course of Olaparib (10 μM) induced TBK-1 and IRF-3 phosphorylation between control and GRB2-KO HeLa cells. **d** GRB2 re-expression in GRB2-KO HeLa cells suppressed Olaparib (10 μM, 48 h) induced TBK-1 and IRF-3 phosphorylation. **e** Representative immunofluorescence images showing olaparib (10 μM, 48 h) treatment induces nuclear accumulation of phosphorylated IRF-3 (pIRF-3) in GRB2-KO HeLa cells. Scale bar 10 μm. **f** The increased TBK-1 and IRF-3 phosphorylation observed in GRB2-KO HeLa cells are independent of RAS (PD184352; left panels) or Akt (MK2206; right panels) signaling. **g** The increased TBK-1 and IRF-3 phosphorylation in GRB2-KO HeLa cells are MRE11 dependent, n = 3. **h** STING knockdown is sufficient to abrogate Olaparib (10 μM) induced TBK-1 and IRF-3 phosphorylation in GRB2-KO HeLa cells. n = 3. **i** The increased TBK-1 and IRF-3 phosphorylation in GRB2-KO HeLa cells are STING dependent, n = 3. **i–k** qRT-PCR showing Olaparib induce increased level of INF-B, CCL5 and CXCL-10 mRNA in HeLa cells with ±SD, n = 3. **l** Quantitation of cytokines array results from two independent experiments showing upregulated inflammatory cytokine released in culture medium with olaparib treatment, n = 2. **m** Recruitment of cytotoxic CD8 + T-lymphocytes (CTL) isolated from PBMC to GRB2-KO HeLa cells in response to Olaparib treatment, n = 3, ±SD are shown. **n** siRNA mediated STING knockdown in GRB2-KO HeLa cells abrogate PBMC isolated CD8 + CTL recruitment, n = 3, errors are in ±SD. The significance was analyzed by two-sided Student's t test. *P ≤ 0.05, **P ≤ 0.01, ***P ≤ 0.001, and ****P ≤ 0.0001; NS not significant. For (**b–n**), source data are provided as a Source Data file.

mRNA upregulation correlated with secretion of these chemoattractants, we used a human inflammation antibody array to measure cytokines into the extracellular environment. In response to PARPi treatment, GRB2-KO cells secreted a higher concentration of chemoattractant cytokines for T-cells (CCL5), myeloid and dendritic cells (CCL2), T-cells and NK-cells (CXCL 10) (Fig. 4l and Supplementary Fig. 8d). We also observed a significant increase in the secretion of PDGF-BB, a known mitogen and a chemoattractant[51].

We therefore investigated if the released cytokines from GRB2-KO cells are sufficient to attract cytotoxic CD8+ cytotoxic T-cells. Peripheral blood mononuclear cells (PBMC) were isolated and tested in a cell-migration recruitment assay. Only GRB2-KO cells that were treated with PARPi showed a significant increase in cytotoxic T-cell infiltration compared to untreated and the control cells with normal level of GRB2 (Fig. 4m). Knockdown of STING in GRB2-KO cells abrogated CD8+ cytotoxic T-cells infiltration (Fig. 4n).

These collective and complementary data support a key role of nuclear GRB2 in the DNA RF stress response. In cultured HeLa and HAP-1 cells under replication stress induced by PARPi, GRB2 deficiency causes replication fork defects leading to genomic instability and cytosolic DNA accumulation. This then activates cGAS/STING leading to production of chemoattractant cytokines, which in turn recruit cytotoxic T-cells.

### PARPi averts tumor progression in a GRB2-depleted ovarian cancer model

Having established functional and mechanistic roles for GRB2 during RF stress that restrict inflammatory cytokine release, we directly investigated immune surveillance of PARPi-treated cancer cells with low GRB2 in a whole animal setting. Our prior cancer database analyses suggested that ovarian cancer with low GRB2 expression correlated with increased cytotoxic T-cell infiltration[52]. We therefore tested this notion experimentally by using metastatic ovarian cancer cell-line ID8. We generated Grb2-KD ID8 mouse cells and found that PARPi also induced TBK1 phosphorylation (Supplementary Fig. 8e–g). In preparation for oral dosing, we also compared TBK1 phosphorylation level between Olaparib and Talazoparib, where Talazoparib induced higher TBK1 response with lower dose (Supplementary Fig. 8f, g) establishing it as a suitable substitution for Olaparib in mice. Grb2-KD mice receiving Talazoparib showed a reduced cancer burden (Fig. 5a and Supplementary Fig. 9a). This was recapitulated in our tumor volume measurements based on total flux (photons per second) measurements. Grb2-KD cells treated with Talazoparib also showed a significantly slower tumor growth rate (Fig. 5b). Furthermore, Grb2-KD mice treated with Talazoparib survived significantly longer than those treated with PBS or the control cells irrespective of treatment conditions (Fig. 5c).

A major complication associated with metastatic ovarian cancer is ascites buildup, which is often associated with advanced disease[53]. Grb2-KD cells injected mice treated with PARPi showed significantly lower body weight due to reduced ascites volume compared to the control mice and to vehicle treated groups (Fig. 5d, and Supplementary Fig. 9b, c). The inflammatory cytokine interleukin 12 (IL-12) is often associated with cytotoxic T-cell and NK cell activation[54], so we measured the IL-12 and IL-12 p40 level in the ascites fluid. We found significant elevation in mice bearing Grb2-KD cells and treated with PARPi indicative of high immune cell activities (Fig. 5e, f). These collective data show that low GRB2 levels enable cGAS/STING activation and immune detection of cancer cells.

## Discussion

As a cytoplasmic intracellular signaling adapter protein, GRB2 links RTKs to the Ras-MAPK cascade to promote cell proliferation[1,4,5,55,56]. However, how nuclear GRB2 can moonlight in the nucleus to impact the replication stress response has been unappreciated. Pulldown followed by mass spectrometry initially identified PCNA complexed with GRB2[11]. Our in vitro measurements confirmed a direct binding with a biologically relevant Kd. We identified a PCNA interacting peptide (PIP) motif on GRB2 as a binding surface plus a second mode of interaction between phosphorylated PCNA (Y211) and GRB2-SH2 domain. In-cell binding measurements with FLIM showed GRB2-PCNA at DNA replication sites, but only in a subset of foci. Our iPOND data provided a context to the FLIM measurements and suggest GRB2-PCNA interaction predominantly occurs on progressing replication forks but not on stalled forks. FLIM imaging of GRB2-PCNA interaction therefore may provide an enabling new technique to delineate the level of stalled versus progressing replication forks in cells.

Having found GRB2 enrichment at replication forks, we investigated this further and discovered that GRB2 acts in response to HU and PARPi by stabilizing RAD51 on DNA to reduce both cytoplasmic DNA accumulation and cGAS/STING activation. Thus, in the context of RF stress, GRB2 depletion promotes cGAS/STING activation that translated into the production of inflammatory cytokines and recruitment of cytotoxic T-cells. Moreover, in the immune competent mouse model, PARPi treatment of cancer cells with low-GRB2 leads to an enhanced targeted destruction tumor cells by the host immune system compared to PARPi alone. This combination of GRB2 depletion and PARPi treatment enabled longer survival and less cancer-associated complications.

Mechanistically, we showed that GRB2 inhibits RAD51 ATPase activity, which restricts MRE11-mediated fork degradation during replication stress. In the absence of GRB2, excessive fork degradation by MRE11 produces cytoplasmic DNA and cGAS/STING activation. PARPi-induced cGAS/STING activation in GRB2-KO cells can therefore be rescued by MRE11 depletion. However, MRE11 inhibition in GRB2-KO cells paradoxically shows cGAS/STING activation. This finding uncovers a possible neomorphic relationship of MRE11i and gene depletion in the context of GRB2 expression. In the absence of GRB2, the inhibited MRE11 protein may promote other incision activities than

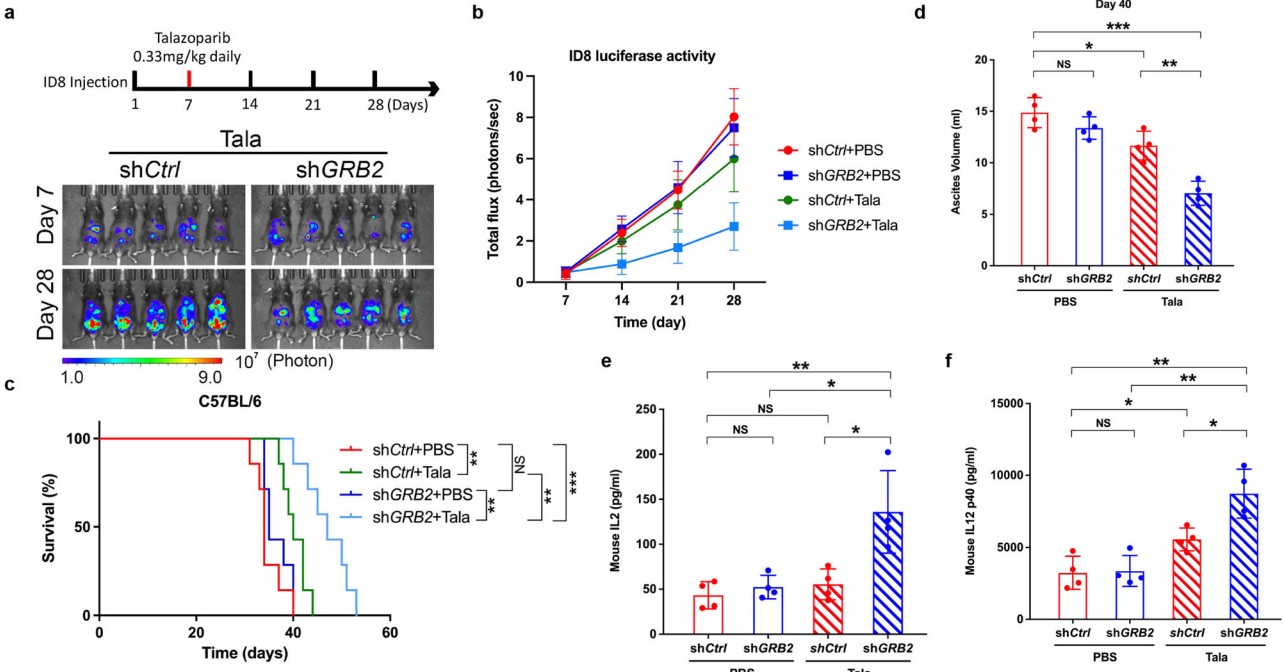

**Fig. 5 | GRB2 depletion sensitizes cells to PARPi. a** A schematic experimental overview for monitoring cancer growth with luciferase imaging. Below, Representative images of luciferase expression ID8 cells detected on day-7 and day-28. Talazoparib (0.33 mg/kg) treated control and GRB2-KD mice are only shown. **b** Quantitative luciferase signal detection using IVIS imaging for $n = 5$ mice per group up to 28 days, the mean with ±SD are shown. **c** Kaplan–Myer curve showing survival of mice in the four-treatment group. **d** The measurements of Ascites volume $n = 4$ mice all 4 treatment groups, the mean with ±SD are shown. **e, f** Measurement of mouse IL-2 level and IL-12 in the Ascites fluids of $n = 4$ mice, the mean with ±SD are shown. The significance was analyzed by two-sided Student's $t$ test. *$P \leq 0.05$, **$P \leq 0.01$ and ***$P \leq 0.001$; NS not significant. For (**b–f**), source data are provided as a Source Data file.

those engaged by MRE11 in the presence of GRB2, such as by the replication fork enzyme exonuclease V[57]. In other systems, inactive enzyme binding can enforce such alternative incision processes as does MRE11i for homologous recombination repair[58,59].

In general, we found that GRB2 depletion mirrors BRCA deficiency in loss of replication fork protection and PARPi sensitivity. In an acute cytotoxicity assay, GRB2 KO cells showed 10-fold higher sensitivity to PARPi[11], yet the level of PARPi sensitivity is not at the same level seen in BRCA2-deficient cells[60,61]. Nevertheless, GRB2-depleted cells display many of functional characteristics of the BRCA2-deficient phenotype. Like BRCA2, GRB2 blocks MRE11-mediated degradation and stabilizes RAD51 on DNA at reversed replication forks. Furthermore, BRCA2-depletion in GRB2-KO cells showed no additional replication fork defect implying an epistatic role of GRB2 and BRCA2 at stalled RFs. MRE11 knockdown (MRE11i at a lesser extent) rescued fork protection in GRB2-KO cells. Beside MRE11 regulation[11], here we find GRB2 has roles in RAD51 filament stabilization. In MRE11i-treated cells with normal MRE11 levels, we envisage that the level of GRB2 stabilized RAD51-mediated protection is a determining factor for the level of fork degradation. In MRE11-KD cells, the amount of MRE11 available for nuclease activity becomes a determining factor. Either insufficient MRE11 inhibition or a neomorphic activity for inhibited MRE11 may rebalance toward fork degradation in the absence of normal GRB2-promoted MRE11 regulation and RAD51 fork protection. However, a potential MRE11i promoted multi-protein neomorphic phenotype merits further investigation beyond this report.

Importantly, BRCA germline cancers are a proven genetic paradigm to reveal therapy vulnerabilities and resistance mechanisms central to new clinical cancer trials. While the initial discoveries were implemented on BRCA germline patients with breast or ovarian cancer, it has become evident that BRCA and related molecular pathways (BRCAness) play an important role in diverse cancers[62,63].

Therefore PARPi, which exquisitely target BRCAness, have rapidly expanded in treatment and clinical trials of diverse cancer[64]. Yet both acquired and intrinsic resistance to PARPi limits success. Here we define GRB2 roles in the BRCAness MRE11-RAD51-PARPi axis and its possible consequences for therapy response. In particular, GRB2 stabilization of RAD51 filaments at replication forks is mechanistically distinct from the RAD51C-XRCC3 paralogs that act by capping the RAD51 filament[65]. Instead, GRB2 acts by inhibiting the RAD51 ATPase activity that promotes filament disassembly in the presence of ssDNA. Blocking GRB2 activity may therefore offer a biological and potentially attractive clinical alternative to stabilizing RAD51 filaments by inhibiting the PolQ ATPase stimulation by ssDNA by novobiocin, which has entered cancer clinical trials[66]. Furthermore, GRB2 may also regulate MRE11 by destabilizing the MRE11 dimer, as recently found for DYNLL1[11,67]. Therefore, GRB2 inhibitors may destabilize forks and activate innate immunity by synergistically both reducing RAD51-mediated fork protection and increasing MRE11-mediated fork degradation. These collective data thus uncover new GRB2 and BRCAness biology that may help clinical researchers to better target and more fully leverage so far promising PARPi treatment strategies and therefore to develop new combination treatment strategies engaging immune therapy.

## Methods
### Reagents
Anti-PCNA (2586), Anti-STING (13647), Anti-MAVS (3993) anti-cGAS (15102), Anti-ERK1/2 (4370), Anti-ERK1/2 (4695), Anti-p-Akt (4060), Anti-Akt (4685), anti-IRF3 (11904), anti-p-IRF3 (29047), anti-TBK1 (3504), anti-p-TBK1 (5483), anti-MRE11 (4847), anti-Histone H3 (4499) and anti-β-Actin (3700), Anti-Flag-tag (14793), Anti-GAPDH (2118), BRCA2 (10741), SMARCAL1 (44717), HTLF (43345) antibodies were ordered from Cell Signaling Technology. Anti-GRB2 (C-23) (sc-255), was purchased from Santa Cruz Biotechnology. Anti-MRE11 (ab214) were

purchased from abcam. Anti-BrdU (347580) was from BD Biosciences. Anti-Rad51 (GTX100469) and ZRANB3 (GTX66576) were ordered from GeneTex. CD8 (301014), Mouse IL2, Mouse IL-12 p40 (ELISA kit 431004) were from Biolegend Olaparib, MK2206, PD184352 and BMN673 (S7048, Talazoparib) was ordered from Selleck Chemicals. CellTiter-Glo was purchased from Promega. PicoGreen was from ThermoFisher and Hydroxyurea, Edu and Thymidine were purchased from Sigma. RFP-GRB2 as previously described[68], mEGFP-PCNA-19-SV40NLS-4 and mEmerald-PCNA-19 without NLS were a gift from Michael Davidson (Addgene plasmid # 56469 and # 54221), PCNA-VHH-TagRFP chromobody was purchased from Chromotek, Flag-BRC4 and RAD51 K133R plasmids were as previously reported[37]. Human Inflammation Array C3 (AAH-INF-3) was purchased from RayBiotech and performed as instructed by the manufacturer. shRNA and siRNA: siMRE11 (L-009271-00-0005), siBRCA2 (L-003462-00-0005), siSMARCAL1 (L-013058-00-0005), siHLTF (L-006448-00-0005), siZRANB3 (L-010025-01-0005), siRAD51 (L-003530-00-0005), siMAVS (L-024237-00-0005), siSTING (L-024333-00-0005) and SiCtrl (D-001810-01-05) were purchased from Dharmacon. The following DNA sequences were purchased from Integrated DNA Technologies. 90nt-ssDNA (5′-CGGGTGTCGGGGCT GGCTTAACTATGCGGCATCAGA GCAGATTGTACTGAGAGTGCACCAT ATGCGGTGTGAAATACCGCACAGATGCGT-3′) and 32nt-dsDNA forward, (5′ –ATATGCGGTGTGAAATACCGCACAGATGCGT-3′) reverse (5′-ACGCATCTGTGCGGTATTTCACACCGCATATG-3′_Cy5). Underlined the homologous region in the 90nt-ssDNA. Synthetic GRB2 PIP motif peptide GASHGQTGMFPRNYVT was purchased from GenScript.

## Cell culture

HeLa cells were ordered from ATCC and maintained in RPMI 1640 medium, and HAP1 cells were purchased from Horizon Discovery and maintained in Iscove's Modified Dulbecco's medium. The ID8 mouse ovarian cancer cells were kindly provided by Dr. Guang Peng's laboratory (MD Anderson Cancer Center). HeLa GRB2 deficient cells and HAP1 GRB2 deficient cells were generated by using CRISPR/Cas9 system as described previously[11]. ID8 GRB2 stably knocked-down cells were generated by using GRB2 shRNAs which were purchased from Dharmacon (RMM4431-200333332, RMM4431-200335970, RMM4431-200399380, RMM4431-200404712). Media were supplemented with 10% fetal bovine serum (Lonza) and 1% antibiotic/anti-mycotic (Lonza) except The ID8 cells were maintained in DMEM supplemented with 4% fetal bovine serum (Lonza), 5 µg/mL insulin, 1% penicillin/streptomycin, 5 µg/mL transferrin, and 5 ng/mL sodium selenite. Cells were incubated at 37 °C in a humidified incubator with 5% CO2. Mycoplasma testing of these cell lines has confirmed negative results.

## Western blots

Cells were washed with PBS and lysed with RIPA buffer (20 mM Tris-HCl [pH 7.5], 150 mM NaCl, 1 mM EGTA, 1% IGPAL, 1 mM $Na_2EDTA$, 1% sodium deoxycholate, 1 mM β-glycerophosphate, 2.5 mM sodium pyrophosphate, 1 mM Na3VO4, 1 µg/ml leupeptin) supplemented with Protease Inhibitor Cocktail Set III (EMD Millipore) to obtain total proteins. The proteins were separated by SDS gel electrophoresis. Membranes were blocked in 5% milk (dissolved in 0.1% Tween 20 TBS buffer) for 1 h at room temperature. Membranes were then incubated with indicated antibodies at 4 °C overnight. Subsequently, membranes were washed with TBS-T and incubated with secondary antibody in TBS-T/5% milk. Membranes were washed in TBS-T, and signals were detected by Enhanced Chemiluminescence.

## Protein expression and purification

Expression and purification of [WT]GRB2, K109R, K109A and GRB2-SH2 domain from bacteria have been described previously[11,69]. The RAD51 expression was performed as described previously[70].

## Immunofluorescence and Fluorescence lifetime imaging microscopy

Designated cells were grown on coverslips and fixed with 4% (w/vol) paraformaldehyde (pH 8.0) with or without pre-extraction and washed four times with PBS (pH 8.0). After permeabilization with 0.5% Triton X-100 on ice for 5 min, cells were washed with PBS and incubated with blocking buffer (PBS, 3% BSA, 5% FBS and 0.05% Triton X-100) for 2 h at room temperature or overnight at 4 °C. Following a further three washes with PBS, cells were incubated with primary antibody overnight in PBS, 3% BSA and 0.05% Triton X-100. Cells were then washed five times with PBS and incubated with the fluorescent conjugated secondary antibody for 2–3 h. Following another five times PBS wash, coverslips were mounted onto a slide with ProLong antifade mounting medium containing DAPI (Invitrogen). For pre-extraction before fixation, cells were treated with cytoskeleton buffer (10 mM PIPES [pH 6.8], 100 mM NaCl, 300 mM sucrose, 3 mM $MgCl_2$, 1 mM EGTA and 0.5% Triton X-100) for 5 min on ice and followed by washing with stripping buffer (10 mM Tris-HCl [pH 7.4], 10 mM NaCl, 3 mM $MgCl_2$, 1% Tween 20 and 0.25% sodium deoxycholate) for 5 min on ice. Cells were imaged using either a Leica SP5 II or Zeiss LSM710 confocal microscope. Nuclear signals of staining were further analyzed by ImageJ. Fluorescence lifetime imaging microscopy was performed as described previously[12,68,69].

## PicoGreen staining

PicoGreen staining was performed by using Quant-iT Pico-Green dsDNA Reagent (Thermo Fisher Scientific). Briefly, cells were grown on coverslips and PicoGreen was diluted into cell culture medium at the concentration of 3 µL/mL. The cells were incubated in the presence of PicoGreen at 37 °C for 1 h. Then the cells were washed with PBS and fixed for confocal microscopy with DAPI counterstaining.

## Strand exchange and ATPase assay

For strand exchange, GRB2 was serially half diluted from 20 uM to 0.156 uM in 10 mM HEPES (pH 7.5), 40 mM NaCl, 0.2% glycerol, 0.2 mM TCEP, 20 mM $MgCl_2$ and 0.1 mg/ml BSA. RAD51 (0.5uM) and 90nt-ssDNA (400 nM) in same buffer were incubated at room temperature for 15 min before adding to the serially diluted GRB2. The reaction was initiated by the addition of ATP and 32nt-dsDNA_Cy5 to a final concentration of 0.5 mM and 200 nM respectively. Following a further 10 min incubation at room temperature, the product were analyzed by electrophoresis in 4–20% polyacrylamide gels and quantified as described[71]. The ATPase activity was measured using ADP-Glo reagent (Promega) except the reaction was performed in 10 mM HEPES (pH 7.5), 40 mM NaCl, 0.2% glycerol, 1 mM DTT, 20 mM $MgCl_2$ and 0.1 mg/ml BSA containing 0.5 uM RAD51 and 0.5 uM 32nt ssDNA. The reaction was initiated by addition of 0.5 uM final concentration ATP. Following 2 h at RT or overnight at 4 C incubation, reaction was stopped and developed according to the manufacture instruction. Two hours and overnight produced an identical ATPase inhibition trend but the amplitude of response was better with for overnight reaction.

## Quantitative RT-PCR

Total RNA was extracted from the indicated cells using RNeasy Mini kit (QIAGEN) according to the manufacturer's instructions. cDNA was synthesized for qPCR using iScript™ cDNA synthesis kit (BioRad Laboratories). Real-time PCR reaction mix was prepared by mixing cDNA, primers, and 2 × SYBR green master mix (1708884, Bio-Rad Laboratories) and adding H2O to the 20 µL. The reaction mix was run in an ABI-VIIA7 RealTime PCR Machine (Applied Biosystems). The quantitative PCR analysis was carried out in triplicate with the following primer sets: human CCL5 (forward: 5′-TGCCCA-CATCAAGGAGTATTT-3′; reverse: 5′-CTTTCGGGTGACAAAGACG-3′), human CXCL10 (forward: 5′-GGCCATCAAGAATTTACTGAAAGCA-3′; reverse: 5′- TCTGTGTGGTCCATCCTTGGAA-3′), and human β-Actin

(forward: 5′-GAGCACAGAGCCTCGCCTTT-3′; reverse: 5′-TCATCATC-CATGGTGAGCTG-3′).

## iPOND assay

iPOND assays were performed as previously described[17]. Briefly, cells were treated with the following conditions−10 µM EdU for 15 min, 10 µM EdU followed by 15 µM Thymidine for 1 hr and 10 µM EdU followed by 2 mM HU for 1 hr. Cells were subsequently fixed in 1% formaldehyde solution, quenched with glycine, permeabilized with 0.25% Triton X-100 and clicked with biotin azide. Cell pellets were lysed using 1% SDS in 50 mM Tris-HCl (pH 8) and pull down was performed for 2 hr in 4 °C using 50 µl/sample Streptavidin beads (millipore). Beads were subsequently washed once with 1 ml lysis buffer (5 min), 1 ml low-salt buffer (1% Triton X-100, 20 mM Tris [pH 8.0], 2 mM EDTA, 150 mM NaCl; 5 min), 1 ml high-salt buffer (1% Triton X-100, 20 mM Tris [pH 8.0], 2 mM EDTA, 500 mM NaCl; minutes) and finally twice with 1 ml lysis buffer (5 min). Washed beads were resuspended in 30 µl of 2X Laemmli buffer (BioRad), heated at 100 °C for 10 min and proceeded for immunoblotting.

## In vivo mouse models

All experimental animal protocols described here was approved by the MD Anderson IACUC committee and complied with all ethical guidelines and regulations. Mice were randomly assigned to four groups, 10 mice per group. $5 \times 10^6$ luciferase-labeled ID8 cells (Control; *Grb2*-KD #4) were injected into the peritoneal cavity of BLAB/c mice (female, 6−8 weeks old, CRL/NCI). One week after tumor cell-injection, mice were randomized into 2 treatment groups for each cell lines: half of the control and GRB2-KD#4 cells mice received PBS while the others received Talazoparib (0.33 mg/kg) daily by oral gavage. Tumor progression was monitored once a week by using a Xenogen IVIS Spectrum in vivo bioluminescence imaging system (Small Animal Imaging Facility, MD Anderson Cancer Center). Treatment continued for 4 weeks or continued until the criteria for euthanasia is met. Tumor volume was determined on the basis of the total flux (photons per second). Mice reaching an endpoint requiring euthanasia by IACUC guidelines or weighing more than 32 grams as a result of tumor growth and/or ascites were euthanized. Kaplan-Meier survival curves were calculated for each group and significance calculated by log-rank Mantel-Cox test.

## T cell recruitment assay

Human blood was provided by MD Andersson blood center. Peripheral blood mononuclear cells (PBMCs) were purified with Ficoll (SigmaAldrich) by density gradient separation.

WT and GRB2 KO HeLa cells were pretreated with 20 µM Olaparib for 48 h were seeded onto 24-well plates, and PBMCs were then added on 3-µm hanging cell culture inserts (Millicell, No. MCMP24H48) placed in a 24-well plate. After overnight incubation, cells that migrated to the 24 wells were collected, counted, and stained with fixable viability dye (Invitrogen, No. 65-0863-14) followed by APC/Cyanine7 anti-mouse CD45, FITC anti-CD3, PerCP/Cyanine5.5 anti-CD8 antibodies and analyzed by FACS Canto II flow cytometer.

## SIRF assay

Enrichment of proteins on nascent DNA at stalled replication forks was validated by the quantitative in situ analysis of protein interactions at DNA replication forks (SIRF) assay. Cells grown on coverslips were pre-extracted for 5 min on ice and fixed with 4% paraformaldehyde, and permeabilized with 0.5% Triton X-100. In situ proximity ligation assay (PLA) was performed using Duolink PLA technology (Sigma-Aldrich) according to the manufacturer's instructions. In brief, coverslips were blocked in 2% BSA for 30 min at 37 °C and incubated with the respective primary antibodies diluted in 1% BSA for 1 h at RT. Upon washing the coverslips three times in PBS for 5 min, anti-Mouse PLUS

and anti-Rabbit MINUS PLA probes (Sigma-Aldrich) were coupled to the primary antibodies for 1 h at 37 °C. Coverslips were washed three times in Wash Buffer A (0.01 M Tris, 0.15 M NaCl and 0.05% Tween 20) for 5 min before PLA probes were ligated for 30 min at 37 °C. After three steps wash in Wash Buffer A for 5 min, amplification was carried out using the 'Duolink In Situ Detection Reagents Green' (Sigma-Aldrich) at 37 °C for 100 min. Coverslips were then washed twice in Wash Buffer B (0.2 M Tris and 0.1 M NaCl) for 10 min and once in 0.01x Wash Buffer B for 1 min. Finally, coverslips were mounted with Pro-Long Gold Antifade Mounting with DAPI (Invitrogen), sealed and imaged on a Leica DMI 6000 fluorescence microscope at 63X magnification.

## DNA fiber assay

DNA fiber assay was carried out as described previously[72]. In brief, approximately $1 \times 10^5$ cells were plated in each well of a six-well plate. Cells were pulse-labeled with 250 µM IdU for 30 min, and then pulsed with 100 µM CldU for another 30 min. After labeling and treatment, cells were harvested and resuspended in 200 µl of PBS. Cell lysis mixture (2.5 µl cell suspension mixed with 7.5 µl of lysis buffer (0.5% sodium dodecyl sulfate, 200 mM Tris-HCl [pH 7.4], 50 mM EDTA)) was dropped on the top of a microscope slide, which was later inclined at 45° to allow the suspension spreading slowly down the glass. Once dried, DNA spreads were fixed by incubation for 5 min in a 3:1 solution of methanol-acetic acid. The slides were dried and placed at 4 °C overnight. DNA was denatured with 2.5 N HCl for 30 min. The slides were rinsed several times in PBS and incubated with the following antibodies: mouse anti-BrdU fluorescein isothiocyanate (BD Biosciences) and rat anti-CldU (Abcam) diluted in 1% BSA. After incubation in a humid chamber for 1 h at 37 °C, slides were washed three times, each time for 5 min in PBS containing 0.1% Triton X-100. The slides were incubated with secondary fluorescent antibodies Alexa anti-mouse 488 and Alexa anti-rat 555 (Invitrogen) diluted in 1% BSA for 1 h at 37 °C. Slides were washed three times for 5 min in PBS-0.1% Triton X-100 and mounted with ProLong Gold Antifade Mounting with DAPI (Invitrogen). Pictures were acquired using a Nikon 80 microscope with a 60× lens and analyzed using ImageJ software. Statistics were calculated using Prism software.

## Microscale thermophoresis

The MST method has been described in detail elsewhere[73]. The Kd values were measured using the Monolith NT.115 system (Nano Temper). Proteins were fluorescently labeled with Atto 488 according to the manufacturer's protocol. Labeling efficiency was determined to be 1:1 (protein: dye) by measuring the absorbance at 280 and 488 nm. A solution of peptides or proteins in 0.01 M HEPES (pH 7.4), 0.15 M NaCl and 0.005% v/v Surfactant P20 was serially diluted, typically from about 100 µM to 30 nM in the presence of 100 nM labeled protein. The samples were loaded into silica capillaries (Polymicro Technologies) after incubation at room temperature for 15 min. Measurements were performed at 22 °C using 20% LED power and 40% IR-laser power. Measurements were also carried out using 20% and 60% IR-laser power for comparison. Data analyses were performed using Nano Temper Analysis software using the Kd curve fitting function. Raw data were exported, and fitting curves were generated using Prism 8 (GraphPad Software).

## Statistical analysis

All statistical analyses were performed using GraphPad Prism 8 utilizing multiple *t*-test function.

## Reporting summary

Further information on research design is available in the Nature Portfolio Reporting Summary linked to this article.

## Data availability

All data supporting the findings of this study are available within the paper and its Supplementary Information. Source data are provided with this paper.

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

## Acknowledgements

This research was supported by the University Cancer Foundation via the Institutional Research Grant program at The University of Texas MD Anderson Cancer Center (to Z.A.); by Cancer Prevention Research Institute of Texas (CPRIT) grant RP180813 and by National Institutes of Health (NIH) grants R01 CA200231, P01 CA092584, R35 CA220430, and 1S10OD012304-01. J.A.T.'s efforts are also supported by a Robert A. Welch Chemistry Chair (G-0010). This work also supported by NIH grant 5R01CA181663 to G.P., CA248088 to B.W. K.S. and S.R. were supported by NIEHS 1R01ES029680 and Cancer Prevention and Research Institute of Texas RP180463. K.S. is a CPRIT Scholar in Cancer Biology and a Rita Allen Foundation Fellow. For generating Supplementary Fig. 7j,k,l we acknowledge Z. Y.'s effort supported by Natural Science Foundation of Zhejiang Province of China (Grant No. LTGY23H160018) and Zhejiang Medical and Health Science and Technology Program (Grant No. 2024KY789).

## Author contributions

Z.A. and J.A.T. directed the research; Z.A., J.A.T., and Z.Y. conceived the research plan; Z.Y., S.X., Y.S. and Z.A. performed the experiments; Z.Y., S.X., Y.S., X.C., Y.Z., S.R., and Z.A. designed experiments and analyzed results; S.N., M.L., and T.L., purified proteins and performed analysis; K.S., G.P., D.Y., and B.W. provided critical reagents and resources; and Z.Y., J.A.T., and Z.A. wrote the paper.

## Competing interests

The authors declare no competing interests.
