## [Peer Review File · Nature Communications]

GRB2 stabilizes RAD51 at reversed replication forks
suppressing genomic instability and innate immunity against
cancerREVIEWER COMMENTS

Reviewer #1 (Remarks to the Author):

In this manuscript by Ye et. al., the authors show that GRB2 associates with the replication fork and protects DNA from MRE11 degradation. They also show that GRB2 deficiency results in the activation of the cGAS/STING pathway, leading to increased chemokine production. Finally, they found that GRB2 deficiency enhances PARPi responses in a mouse model of ovarian cancer. Overall, the data in the manuscript are performed well and convincing that GRB2 plays a critical role in DNA repair. This provides novel insight into the potential nuclear function of GRB2, which is mostly unknown. There are minor changes or additions that would strengthen the paper.

The data shown in Extended Data Fig 2 are puzzling. It appears that the majority of the PCNA is in the nucleus when the cells express RFP alone. However, when the GFP-PCNA is expressed with RFP-GRB2 it appears that either 1) PCNA is found equally in the nucleus and cytoplasm or 2) RFP-GRB2 is found exclusively in the nucleus. From the images, it appears that option 2 is the most likely. If this is true, how do the authors explain the differences in PCNA expression? Also, this would suggest that GRB2 is interacting with PCNA outside of the nucleus, an observation that is supported by data in Fig 1b that shows that PCNA and GRB2 recruitment to the DNA replication fork (RF) is inversely proportional. This data appears to argue against their model that this interaction is occurring at the RF. This would only pertain to the GRB2-PCNA interaction, since the authors present compelling evidence that GRB2 is recruited to RF.

There is no discussion of the number of replicates in the in vitro binding experiments shown in Fig 3A. Additionally, is it possible to show the error in the affinity measurements and provide statistical analysis of this data.

There is no discussion of the number of replicates performed for the immunoblotting experiments in Fig 4 and Extended Data Fig 8. The authors should also quantify and compile data from the multiple replicates and perform statistical analysis on these data, especially in light of the fact that they are comparing multiple treatments.

The discussion is very short for the amount of data presented in the paper. It would be helpful to have further discussion, especially of a tentative model for how GRB2 controls DNA repair responses.

The callout on line 201 is incorrect. Should read Extended Data Fig. 6g.

Reviewer #2 (Remarks to the Author):

Ye et al build upon prior work demonstrating a nuclear role for GRB2 in HR-mediated stability of stalled replication forks. In the current study, the authors demonstrate that GRB2 is necessary for end protection of stalled replication forks from degradation by the Mre11 nuclease. They demonstrate that this GRB2 effect is epistatic to BRCA2, and entails inhibition of Rad51 ATPase activity to stabilize Rad51 at stalled forks. Depletion of GRB2 in PARP inhibitor treated cells stimulates Mre11-dependent accumulation of cytosolic DNA and cGAS/STING-dependent activation of interferon signaling. Overall, the molecular biology studies are well controlled and the results support the interpretation. However, there are a few aspects of the cellular studies that could be further evaluated to strengthen the link to cancer relevance.

Major Critiques:

1) Does GRB2 knockout (or preferably the K109R mutant) result in PARP inhibitor sensitivity to similar levels as BRCA2 mutation? Comparison of these effects in an isogenic cell system would be helpful to

understand to what degree GRB2 is necessary for BRCA2 functions in the setting of PARP inhibitor sensitivity.

2) Does overexpression of Rad51-K133R reduce PARPi sensitivity of GRB2 knockout (or preferably the K109R mutant) cells? Ideally, this may require doxycycline-induced expression of Rad51-K133R to find a level that is not toxic to cell growth. This would clarify the extent to which PARPi sensitivity is due to the proposed mechanism of Rad51 filament stability versus other defects in completion of HR.

3) The role of Mre11 in cGAS activation is based on an Mre11 knockdown experiment (Figure 4g). Does Mirin treatment or expression of a nuclease-dead Mre11 recapitulate these effects? Prior work has indicated that Mre11 mutant/deficient cells are also sensitive to PARP inhibitors, so the assertion here that Mre11 nuclease activity is the basis for sensitivity to PARP inhibitor is a bit confusing. It could be that degradation of unprotected stalled forks is Mre11 nuclease-dependent, but blocking Mre11 nuclease is not sufficient to restore PARPi resistance because fork restart pathways are still impaired. Also, a prior study has indicated a positive role for Mre11 in STING activation, which may not necessarily require its nuclease activity (PMID 23388631). Thus further clarification of the role of Mre11 in both PARPi sensitivity and STING-dependent inflammatory signaling is warranted.

4) Examination of Grb2 in human cancers through the cBioportal indicates relatively infrequent genetic alterations. Could the authors comment on the clinical relevance of their findings in the context of human cancers in the discussion? Are they suggesting that transcriptional silencing of the gene is contributing to responses in BRCA-wild type tumors? Or is there another mechanism of suppressing GRB2 functions in replication fork stability?

Minor points:

1) TBK1 is often misspelled as TKB1.

2) The dosage of Olaparib and Talazoparib used in Figures 4 -5 should be indicated in the figure legends.

Detailed Responses to Reviewers

We thank the reviewers for insightful comments and suggestions. We have taken their suggestions and done additional experiments to address them. New experiments yielded results that further solidified our conclusions. We trust that the reviewers will be pleased with the added new work. All experimental results are presented in the Extended Data Figures. The main text was modified to incorporate the added data. All changes in the text are highlighted in red. We made small adjustment to the title and the current affiliation of Dr. Zu Ye is also added.

Extended data rearrangements.

1. New data for extended data Fig. 2c, old 2c is now 2d. (PCNA)
2. Quantifications of pTBK/tTBK and pIRF3/tIRF3 from three independent experiments and from two different cell-lines are added in Extended data Fig. 7c-k and Extended data Fig.8 a, c and g.
3. New data on Extended Fig. 7j,k. (MRE11i)
4. New data in Extended data Fig. 6h,i (^{K133R}RAD51 expression)

Reviewer #1 (Remarks to the Author):

In this manuscript by Ye et. al., the authors show that GRB2 associates with the replication fork and protects DNA from MRE11 degradation. They also show that GRB2 deficiency results in the activation of the cGAS/STING pathway, leading to increased chemokine production. Finally, they found that GRB2 deficiency enhances PARPi responses in a mouse model of ovarian cancer. Overall, the data in the manuscript are performed well and convincing that GRB2 plays a critical role in DNA repair. This provides novel insight into the potential nuclear function of GRB2, which is mostly unknown. There are minor changes or additions that would strengthen the paper.

The data shown in Extended Data Fig 2 are puzzling. It appears that the majority of the PCNA is in the nucleus when the cells express RFP alone. However, when the GFP-PCNA is expressed with RFP-GRB2 it appears that either 1) PCNA is found equally in the nucleus and cytoplasm or 2) RFP-GRB2 is found exclusively in the nucleus. From the images, it appears that option 2 is the most likely. If this is true, how do the authors explain the differences in PCNA expression? Also, this would suggest that GRB2 is interacting with PCNA outside of the nucleus, an observation that is supported by data in Fig 1b that shows that PCNA and GRB2 recruitment to the DNA replication fork (RF) is inversely proportional. This data appears to argue against their model that this interaction is occurring at the RF. This would only pertain to the GRB2-PCNA interaction, since the authors present compelling evidence that GRB2 is recruited to RF.

We have performed additional experiments altering the tag position on GRB2, plus PCNA without the NLS and a fluorescence labelled PCNA-chromobody that detects endogenous PCNA. For GRB2, n-terminal RFP-tagged GRB2 showed a diffused cellular distribution while C-terminally GFP-tagged GRB2 showed enriched nuclear localization. Similarly, PCNA without a NLS (mEmerald-PCNA) is mostly cytoplasmic. GFP-PCNA used in Fig. 1a despite the 1xNLS, some cytoplasmic localizations were seen. We also tested a TagRFP labelled PCNA-chromobody to detect endogenous PCNA, which showed, some cytoplasmic but predominantly nuclear localization in most cells. Thus, our collective data points to low PCNA in the cytoplasm of the cells and that shown in Fig. 1a is an overexpression artifact independent of GRB2 expression. New data presented in Extended data Fig. 2c and referenced in the main text. Notably, our FLIM method for single photon counting circumvents contrast-based imaging

localization artifacts and accurately measures direct interaction between GRB2 and PCNA inside cells.

There is no discussion of the number of replicates in the in vitro binding experiments shown in Fig 3A. Additionally, is it possible to show the error in the affinity measurements and provide statistical analysis of this data.

Apologies, the experimental replicates are now added to the figure legends. Kd numbers with SD is also added to Fig. 3a.

There is no discussion of the number of replicates performed for the immunoblotting experiments in Fig 4 and Extended Data Fig 8. The authors should also quantify and compile data from the multiple replicates and perform statistical analysis on these data, especially in light of the fact that they are comparing multiple treatments.

Fig. 4 and Extended data Fig. 8 now report the concentration of Olaparib and the duration of treatment. We have also compiled all quantifications of 3x replicates in both HeLa and HAP1 cell-lines and presented them in Extended data Fig. 7c-i & 8a,c and g.

The discussion is very short for the amount of data presented in the paper. It would be helpful to have further discussion, especially of a tentative model for how GRB2 controls DNA repair responses.

Thank you for your suggestions. Additional discussion has been added.

The callout on line 201 is incorrect. Should read Extended Data Fig. 6g.
Changed

Reviewer #2 (Remarks to the Author):

Ye et al build upon prior work demonstrating a nuclear role for GRB2 in HR-mediated stability of stalled replication forks. In the current study, the authors demonstrate that GRB2 is necessary for end protection of stalled replication forks from degradation by the Mre11 nuclease. They demonstrate that this GRB2 effect is epistatic to BRCA2, and entails inhibition of Rad51 ATPase activity to stabilize Rad51 at stalled forks. Depletion of GRB2 in PARP inhibitor treated cells stimulates Mre11-dependent accumulation of cytosolic DNA and cGAS/STING-dependent activation of interferon signaling. Overall, the molecular biology studies are well controlled and the results support the interpretation. However, there are a few aspects of the cellular studies that could be further evaluated to strengthen the link to cancer relevance.

Major Critiques:

1) Does GRB2 knockout (or preferably the K109R mutant) result in PARP inhibitor sensitivity to similar levels as BRCA2 mutation? Comparison of these effects in an isogenic cell system would be helpful to understand to what degree GRB2 is necessary for BRCA2 functions in the setting of PARP inhibitor sensitivity.

We previously reported a 10-fold PARPi sensitivity of GRB2-KO cells over the matched control and reconstitution of WT but not K109R mutant in KO cells reverse this effect (Ye et al 2021, Sci Adv). We observed the hypersensitivity of olaparib in both GRB2-KO and BRCA2 KD cells, however PARPi sensitivity in GRB2-KO cells were not to an extent those seen in BRCA2 deficiency.

We expect that follow up efforts by ourselves and others will further define this relationship. In fact, our ongoing work has identified a previously unknown GRB2-BRCA2 interaction, which is beyond the scope of this manuscript. Through a series of BRCA2 truncation mutants, we identified a previously undiscovered kinase regulated motif and site for GRB2-BRCA2 interaction (see attached data below). Since the GRB2 interaction site falls outside all currently known BRCA2 mutants, our ongoing effort is focused on generating BRCA2 mutants that specifically disrupt GRB2 binding while maintaining all other functions. This investigation in progress is not complete and requires many experiments that would then be published in a future manuscript. Although related, inclusion of this preliminary data would be premature and a diversion to the scope and conclusions of the current manuscript. We note it here for the reviewers as consistent with participation of GRB2 in the BRCA2 network. In this manuscript, we present strong data from years of work showing the unexpected mechanism whereby GRB2 lowers the probability of RAD51 filament dissociation by blocking its ATPase activity and of MRE11 nuclease degradation of stalled replication forks, as consistent with supporting BRCA2 functions.

2) Does overexpression of Rad51-K133R reduce PARPi sensitivity of GRB2 knockout (or preferably the K109R mutant) cells? Ideally, this may require doxycycline-induced expression of Rad51-K133R to find a level that is not toxic to cell growth. This would clarify the extent to which PARPi sensitivity is due to the proposed mechanism of Rad51 filament stability versus other defects in completion of HR.

We thank the reviewer for this suggestion, while the doxycycline-induced system proved challenging, we have performed a K133R dose escalation expression analysis coupled cell-survival assay in HAP1 GRB2-KO cells. Our results show a 2x RAD51 K133R mutant expression in GRB2-KO cells is sufficient to confer PARPi resistance. The data is consistent with our hypothesis that in GRB2 depleted cells, high RAD51 ATPase activity is detrimental for fork protection when exposed to PARPi. These data have now been added to the Extended data Fig. 6h,i.

3) The role of Mre11 in cGAS activation is based on an Mre11 knockdown experiment (Figure 4g). Does Mirin treatment or expression of a nuclease dead Mre11 recapitulate these effects?

To answer this question, we treated control and GRB2-KO cells with MRE11 inhibitor Mirin, PFM01 or PFM39 with and without olaparib and analyzed the level of pTBK1 levels. Results from multiple cell-lines revealed MRE11 inhibition in GRB2-KO did not fully recapitulate those seen with MRE11-KD (Extended data Fig. 7j,k). There was cell-type specific response to MRE11i, In HeLa cells a measurable reduction in pTBK1 control cells when MRE1i was combined with PARPi. Thus, MRE11i may either lack sufficient potency to provide full protection against degradation of an unprotected replication fork without GRB2 being present or MRE11i

promotes alternative activities at forks compared to MRE11-KD in the absence of GRB2. We added this new data in Extended fig. 7j and k and added additional discussion.

Prior work has indicated that Mre11 mutant/deficient cells are also sensitive to PARP inhibitors, so the assertion here that Mre11 nuclease activity is the basis for sensitivity to PARP inhibitor is a bit confusing.

We previously found that GRB2-KO cells are HDR deficient and therefore PARPi sensitive. Reconstitution of GRB2-KO cells with K109R mutant that do not bind MRE11 or K109A that binds MRE11 but cannot be ubiquitinated and therefore cannot release MRE11 at the DSBs maintains PARPi sensitivity. We proposed that GRB2-mediated MRE11 recruitment and subsequent release at the DSB sites is important for the formation of functional MRN complex and deficiency in either cause HDR deficiency (Ye et. al. 2021). Thus, previous work showing MRE11 mutant/deficient cells are PARPi sensitive can also be explained by the inability to form a functional MRN complex.

It could be that degradation of unprotected stalled forks is Mre11 nuclease-dependent, but blocking Mre11 nuclease is not sufficient to restore PARPi resistance because fork restart pathways are still impaired.

Indeed, our collective data point to MRE11 nuclease-dependent degradation of unprotected fork in GRB2 depleted cells and the fork protection defects caused by the instability of the RAD51 filament formation. Here MRE11 nuclease function is a causative effect rather than a functional one. Our new data guided by the reviewer show restoration of RAD51 filament stability with ATP hydrolysis defective mutant reverses PARPi-mediated killing of GRB2-KO cells suggesting restoration of fork protection as the likely mechanism.

Also, a prior study has indicated a positive role for Mre11 in STING activation, which may not necessarily require its nuclease activity (PMID 23388631). Thus, further clarification of the role of Mre11 in both PARPi sensitivity and STING-dependent inflammatory signaling is warranted.

Our new data in the revised manuscript strengthened the conclusion where MRE11 nuclease activity is important for STING activation in GRB2-depleted cells but not in cells with high GRB2. As such, MRE11-dependent cGAS/STING activation closely correlates with relative level of cellular GRB2, which can also be cell-line dependent. Cultured cell-lines show varying degree of GRB2 expression depending on the source stage of the tumor from which the cell-line originated. GRB2 expression exquisitely correlated with malignancy and tumor stage, late-stage and malignant tumors show higher GRB2 compared to matched normal (See Ahmed et. al 2015 Nat. Commun). Kondo et. al. (PMID 23388631) MEF cells and a short 6h 100uM mirin treatment to measure *ifnb1* mRNA induction. The short treatment time and high mirin concentration plus introduction of an exogenous DNA through transfection strongly suggest the induction of MAVS pathway for STING activation, which although was not investigated but is a cellular viral response system. Our extended treatment times and results in Fig. 4h and Extended data Fig. 8a-c shows GRB2 effects STING through cGAS activation and not MAVS. Furthermore, our findings of cGAS/STING activation were independent of exogenous DNA transfection.

4) Examination of Grb2 in human cancers through the cBioportal indicates relatively infrequent genetic alterations. Could the authors comment on the clinical relevance of their findings in the context of human cancers in the discussion? Are they suggesting that transcriptional silencing of the gene is contributing to responses in BRCA-wild type tumors? Or is there another mechanism of suppressing GRB2 functions in replication fork stability?

Compared with normal or those at early stages, GRB2 expression is higher in late-stage aggressive tumors. However, it is not clear how GRB2 expression are regulated during tumorigenesis. Further work beyond the scope of this manuscript will be required to elucidate expression regulations.

GRB2 function can be regulated by its self-association status. Late-stage tumors have a high level of monomeric GRB2 compared to normal or those at an early stage (Ahmed et. al. 2015). Previously three mechanisms for monomerization were defined, 1) low GRB2 concentration <400nM *in vitro*; 2) phosphotyrosine ligand binding to the SH2 domain, and 3) GRB2 phosphophorylation on tyrosine at position 160 (Y160). Thus, the binding of GRB2 to RTK induces dimer dissociation and bound monomer phosphorylation on Y160 by RTK prevents self-association. Notably, monomeric GRB2 at 25kDa is permissible for passive diffuse through nuclear pore while the dimeric (50kDa) would require an active transport.

Thus, a mechanistic sequence can be envisaged where receptor tyrosine kinase activation (e.g. EGFR) at the G0 of the cell-cycle that induces cells to enter cell-cycle, GRB2 phosphorylation by RTKs ensure monomeric state – propagating passive diffusion into nucleus where GRB2 can participate in DNA replication and repair at the later phases of the cell-cycle.

Minor points:

1) TBK1 is often misspelled as TKB1.

Corrected

2) The dosage of Olaparib and Talazoparib used in Figures 4 -5 should be indicated in the figure legends.

Olaparib and Talazoparib doses are now added to respective figure legends.

REVIEWERS' COMMENTS

Reviewer #1 (Remarks to the Author):

The authors have sufficiently addressed all the major comments.

Reviewer #2 (Remarks to the Author):

The revised manuscript has adequately addressed my critiques, and provides compelling evidence for Grb2-mediated stabilization of Rad51 filaments to promote genome stability and suppress innate immune activation by unprotected replication forks.